**Investigation**

# Spatially explicit estimation of recent migration rates in plants using genotypic data

Igor J. Chybicki [ID],[1,]* Juan J. Robledo-Arnuncio [ID][2,]*

[1]Department of Genetics, Kazimierz Wielki University, Chodkiewicza 30, 85064 Bydgoszcz, Poland
[2]Institute of Forest Sciences (ICIFOR-INIA), CSIC, Ctra. De la Coruña km 7.5, 28040 Madrid, Spain

*Corresponding author: Department of Genetics, Kazimierz Wielki University, Chodkiewicza 30, 85064 Bydgoszcz, Poland. Email: igorchy@ukw.edu.pl; *Corresponding author: Institute of Forest Sciences (ICIFOR-INIA), Spanish National Research Council (CSIC), Ctra. De la Coruña km 7.5, 28040 Madrid, Spain. Email: robledo.juan-jose@inia.csic.es

We present a new hierarchical Bayesian method using multilocus genotypes to estimate recent seed and pollen migration rates in a spatially explicit framework that incorporates distance effects separately for each type of dispersal. The method additionally estimates population allelic frequencies, population divergence values, individual inbreeding coefficients, individual maternal and paternal ancestries, and allelic dropout rates. We conduct a numerical simulation analysis that indicates that the method can provide reliable estimates of seed and pollen migration rates and allow accurate inference of spatial effects on migration, at affordable sample sizes (25–50 individuals/population) when population genetic divergence is not low ($F_{ST} \geq 0.05$), or by increasing sampling (to at least 100 individuals/population) under weaker levels of divergence ($F_{ST} = 0.025$). Simulations also show that the accuracy provided by assays with about one thousand unlinked polymorphic SNP loci may approach, for a given sample size, the theoretical maximum achievable under categorical origin discrimination. We apply our method to *Taxus baccata* data, revealing low but significant seed and pollen migration among nearby population remnants during the last generation, with a negative effect of interpopulation distance on migration that was detectable for pollen but not for seeds.

Keywords: seed and pollen dispersal; gene flow; zygotic and gametic migration; isolation by distance; *Taxus baccata*

## Introduction

The rate and scale of migration among populations determines fundamental demographic and genetic processes, including metapopulation dynamics, reproductive and genetic connectivity in fragmented landscapes, adaptive divergence across heterogeneous habitats, colonization, and allochthonous introgression. Due to global change threats, inferring patterns of recent migration under current demographic and environmental conditions, rather than long-term averages under historical ones, is becoming increasingly necessary for understanding the impact of recent landscape transformation on population connectivity, which is crucial for conservation management of natural populations. In the case of plants, ongoing habitat fragmentation is increasing the spatial isolation of populations, with effects that are species-specific and of variable sign, but generally involving altered patterns of interpopulation seed and pollen exchange (Robledo-Arnuncio et al. 2014; Lowe et al. 2015). Moreover, ongoing changes in temperature, precipitation, and wind regimes are expected to alter seed and pollen migration patterns as well, as mediated by variation in the timing and probability of seed abscission and pollen emission (Thompson and Katul 2013; Zhang and Steiner 2022), changes in airborne propagule transport distances and trajectories (Kuparinen et al. 2009; Kling and Ackerly 2020), or phenological and spatial mismatches between plants and their pollinators and dispersers (Gérard et al. 2020). In turn, current patterns of among-population seed and pollen migration affect the expected response of plant populations to climate change, determining not only the speed of suitable habitat tracking but also gene flow across the species range and its ensuing effect on the adaptive potential of populations to their new local climates (Nathan et al. 2011; Kremer et al. 2012; Aguilée et al. 2016; Kling and Ackerly 2020). Monitoring the potential effects of ongoing demographic and environmental changes on plant propagule migration, as well as parameterizing theoretical predictions of range shifts and the interplay between migration and local climatic adaptation, both require empirical measures of the rates of seed and pollen migration and their potential spatial and environmental covariates.

Molecular markers are powerful tools to measure recent migration empirically, since they allow ascertaining the origin of individuals sampled in the field based on their genotypes and those of candidate sources, without requiring forward-tracking or capture-recapture procedures (Manel et al. 2005). Among marker-based field methods employed for plant species, those using parentage assignment can jointly infer spatial patterns of seed and pollen dispersal within a delimited study area, their phenotypic and ecological covariates, and the rate of immigration from (usually unknown) external sources (Ellstrand and Marshall 1985; Adams et al. 1992; Burczyk et al. 2006; Goto et al. 2006; Klein and Oddou-Muratorio 2011; Moran and Clark 2011; Chybicki 2023). Parentage-based methods require exhaustive (Wang 2014)

sampling of reproductive individuals within the study area, however, rendering them generally inefficient to infer recent migration over large spatial scale, unless the density of the target species is very low.

Genetic assignment methods provide a more scalable solution to infer recent among-population migration, by tracing the population (rather than the parental) origin of sampled individuals based on their genotypes and those of random reference samples of individuals from a set of candidate source populations, which becomes possible if the latter are sufficiently genetically divergent among each other (Paetkau *et al.* 1995; Rannala and Mountain 1997; Cornuet *et al.* 1999). Some assignment methods have been explicitly formulated for unbiased estimation of recent migration rates among a set of predefined populations (Pella and Masuda 2001; Wilson and Rannala 2003), with subsequent developments additionally allowing for making inferences about factors affecting observed migration, such as isolation by distance or altitude, under a hierarchical Bayesian framework (Gaggiotti *et al.* 2004; Faubet and Gaggiotti 2008). Plant species exhibit both gametic (pollen) and zygotic (seed) dispersal, which, besides having different demographic and genetic consequences (Lopez *et al.* 2008; Aguilée *et al.* 2013), usually differ in their timing, spatial range, vectors, and environmental determinants (Robledo-Aruncio *et al.* 2014). There are genetic assignment methods available that incorporate this important feature of plants and jointly estimate the rates of recent seed and pollen migration among populations (Robledo-Aruncio 2012; Robledo-Aruncio and Gaggiotti 2017). However, no method has been developed yet to estimate recent seed and pollen dispersal rates along with covariates potentially influencing each type of dispersal.

In this study, we present a new hierarchical Bayesian method using multilocus genotypes to estimate recent seed and pollen migration rates in a spatially explicit framework that incorporates distance effects. Similarly to Faubet and Gaggiotti (2008) approach, spatial data are incorporated through the prior distributions of migration rates, but separately for seed and pollen. The method additionally estimates population allelic frequencies, individual inbreeding coefficients, individual ancestries, and allelic dropout rates. The proposed framework could easily be extended to incorporate additional factors affecting seed and pollen migration. We use simulations to analyze the method's expected estimation errors under contrasting demographic, migration, and sampling assumptions, as well as its ability to detect geographical isolation by distance effects on among-population pairwise migration rates. We illustrate its practical application using previously published real data from European yew.

## Materials and methods
### Demographic model and sampling assumptions

We assume a set of $K$ discrete populations of a diploid plant species that can exchange seed and pollen among each other at variable rates. The observed relative spatial location of populations is defined by a vector $\mathbf{D} = \{d_{ij}\}$, where $d_{ij}$ is the geographical distance between populations $i$ and $j$. Let $\alpha = \{\alpha_{ij}\}$ be the vector of seed migration rates, where $\alpha_{ij}$ is defined as the probability that individuals in population $i$ originate from seeds dispersed from population $j$. And let $\beta = \{\beta_{ij}\}$ be the vector of pollen migration rates, where $\beta_{ij}$ is the probability that seeds produced in population $i$ are the result of pollination by pollen dispersed from population $j$. A total of $N$ individuals is randomly sampled from the populations, and the vector $\mathbf{S} = \{s_i\}$ identifies the population $s_i$ where individual $i$ was sampled. We then have that the $i$-th individual will have dispersed from a mother in population $j$ with probability $\alpha_{s,j}$ and sired by a father in population $k$ with probability $\beta_{jk}$. The model thus reflects the fact that male gametes are first transported via pollen dispersal and then, along with the female gamete, via seed dispersal. Let $\mathbf{o} = \{o_i\}$ indicate the population origin of each sampled individual, so that $o_i = \{j, k\}$ identifies the population source of the $i$-th individual's female and male gametes, respectively. If $\mathbf{o}$ was known and its elements independent, then $\alpha$ and $\beta$ could be estimated based on multinomial likelihood. In general, however, $\mathbf{o}$ is unknown but can be inferred based on genotypic data allowing a probabilistic discrimination of candidate sources.

We assume that sampled individuals are genotyped at $L$ unlinked codominant loci, yielding the vector $\mathbf{G} = \{G_{il}\}$ of observed individual multilocus genotypes, where $G_{il}$ is the observed diploid genotype of individual $i$ at locus $l$. We allow for the possibility of allelic dropout, assuming the vector $\varepsilon = \{\varepsilon_l\}$ that gives the probability that a heterozygous genotype at locus $l$ is mistakenly observed as homozygous for either allele. The unknown population allelic frequencies are given by the vector $\mathbf{p} = \{p_{jla}\}$ that indicates the frequency of allele $a$ at locus $l$ in population $j$. Departures from Hardy–Weinberg equilibrium are allowed by assuming individual-specific inbreeding coefficients $\mathbf{F} = \{F_i\}$.

## Genotypic likelihoods

The likelihood of the data is the probability of the observed genotypes $\mathbf{G}$ given the model parameters:

$$\Pr(\mathbf{G} \mid \mathbf{S}, \ \alpha, \beta, \mathbf{p}, \mathbf{F}, \varepsilon) = \prod_{i=1}^{N} \Pr(G_i \mid \alpha_{s_i}, \beta, \mathbf{p}, F_i, \varepsilon) \tag{1}$$

where

$$\Pr(G_i \mid \alpha_{s_i}, \beta, \mathbf{p}, F_i, \varepsilon) = \prod_{j=1}^{K} \prod_{k=1}^{K} \alpha_{s,j} \beta_{jk} \Pr_{jk}(G_i \mid \mathbf{p}, F_i, \varepsilon) \tag{2}$$

and

$$\Pr_{jk}(G_i \mid \mathbf{p}, F_i, \varepsilon) = \prod_{l=1}^{L} \Pr_{jk}(G_{il} \mid \mathbf{p}_l, F_i, \varepsilon_l) \tag{3}$$

denoting that $g_{il1}$ and $g_{il2}$ are the two observed homologous alleles of individual $i$ at locus $l$ and assuming that identical-by-descent homozygous alleles cannot originate from different populations (i.e. $F_i = 0$ if $j \neq k$):

$\Pr_{jk}(G_{il} \mid \mathbf{p}_l, F_i, \varepsilon_l)$

$$= \begin{cases} F_i p_{jlg_{il1}} + (1 - F_i)(p_{jlg_{il1}}^2 + \varepsilon_l p_{jlg_{il1}}(1 - p_{jlg_{il1}})) & \text{if } g_{il1} = g_{il2} \text{ and } j = k \\ p_{jlg_{il1}} p_{klg_{il1}} + \dfrac{\varepsilon_l}{2}(p_{jlg_{il1}}(1 - p_{klg_{il1}}) + p_{klg_{il1}}(1 - p_{jlg_{il1}})) & \text{if } g_{il1} = g_{il2} \text{ and } j \neq k \\ (1 - F_i)(1 - \varepsilon_l) 2 p_{jlg_{il1}} p_{jlg_{il2}} & \text{if } g_{il1} \neq g_{il2} \text{ and } j = k \\ (1 - \varepsilon_l)(p_{jlg_{il1}} p_{klg_{il2}} + p_{klg_{il1}} p_{jlg_{il2}}) & \text{if } g_{il1} \neq g_{il2} \text{ and } j \neq k \end{cases}$$

$$\tag{4}$$

## Spatially explicit prior distribution of migration rates

Under an isolation by distance framework, seed and pollen migration rates can be expected to depend on the geographical distances between populations ($\mathbf{D}$). We include the expected effect of distance through the separate prior distributions of seed and pollen migration rates. In the case of seed migration rates into population $i$, we consider a Dirichlet prior and assume

$$\alpha_i = \{\alpha_{i1}, \alpha_{i1}, \ldots, \alpha_{iK}\} \sim \text{Dir}(\{\pi_{i1}, \pi_{i1}, \ldots, \pi_{iK}\}, \gamma_\alpha) \quad (5)$$

where $\pi_{ij}$ are expected proportions between 0 and 1 and $\gamma_\alpha$ a dispersion parameter, also between 0 and 1, proportional to the variance of $\alpha_i$ , namely $V(\alpha_{ij}) = \pi_{ij}(1 - \pi_{ij})\gamma_\alpha$. Note that the corresponding standard Dirichlet parameterization would consist of $K$ parameters of the form $a_{ij} = \pi_{ij}(1 - \gamma_\alpha)/\gamma_\alpha$. In turn, we modeled the $\pi_{ij}$ parameters as

$$\pi_{ij} = \begin{cases} \lambda_\alpha + (1 - \lambda_\alpha)\tau_\alpha & \text{if } i = j \\ (1 - \lambda_\alpha)(1 - \tau_\alpha)\rho_{ij} & \text{if } i \neq j \end{cases} \quad (6)$$

where

$$\rho_{ij} = \frac{\exp(-b_\alpha \log(1 + d_{ij}))}{\sum_{k \neq i} \exp(-b_\alpha \log(1 + d_{ik}))} \quad (7)$$

and $\lambda_\alpha$ is a (user-defined) fixed value defining the prior probability of local (within-population) seed dispersal, $\tau_\alpha$ is an isolation parameter, $\rho_{ij}$ is the expected seed migration from population $j$ to population $i$, and $b_\alpha$ is a factor determining the effect of geographical distance on seed migration. As hyperpriors, we assumed uniform densities on (0, 1) for $\gamma_\alpha$ and $\tau_\alpha$ and a Gaussian $b_\alpha \sim \text{Normal}(0, 100)$. An analogous prior structure was assumed for pollen migration rates, with corresponding parameters $\lambda_\beta, \tau_\beta, \gamma_\beta$, and $b_\beta$. This prior parameterization allows formulating a probability density function of seed or pollen migration distances from the source, i.e. a dispersal kernel (Supplement A in Supplementary File 1).

## Other prior distributions

We assumed a model of correlated allele frequencies, where the frequency of alleles in population $j$ at the $l$-th locus ($\mathbf{p}_{jl}$) is determined by the global (ancestral) allele frequencies ($\mathbf{q}_{jl}$) and the genetic divergence between population $j$ and the global gene pool ($F_{ST}^j$). Specifically, we assumed a Dirichlet prior $\mathbf{p}_{jl} \sim \text{Dir}(\theta_{jl})$, where $\theta_{jl} = \mathbf{q}_{jl}(1 - F_{ST}^j)/F_{ST}^j$. As hyperpriors, we considered a flat $\mathbf{q}_{jl} \sim \text{Dir}(1, \ldots, 1)$, a Gaussian $F_{ST}^j \sim \text{Gamma}(\mu_{F_{ST}}, 1)$ truncated at 1, and a uniform density on (0, 1) for the mean $\mu_{F_{ST}}$. In the case of individual inbreeding coefficients, we assumed a Beta prior density $F_i \sim \text{Beta}(\mu_F, \gamma_F)$, where $\mu_F$ is the grand mean inbreeding coefficient and $\gamma_F$ determines the dispersion of individual values around the grand mean. We considered uniform hyperpriors on (0, 1) for $\mu_F$ and $\gamma_F$. Finally, we assumed a Beta prior for the allelic dropout rate at locus $l$, $\varepsilon_l \sim \text{Beta}(0.01, 0.01)$, which is symmetric around 0.5 and has higher density in the neighborhood of the boundaries (0 and 1), being rather uniform elsewhere.

## Posterior distribution of parameters

Given the genotypic data $\mathbf{G}$, the individual sampling locations $\mathbf{S}$, and the population spatial data $\mathbf{D}$ (optional), the joint posterior density over parameter set $\mathbf{\Theta} = (\alpha, \gamma_\alpha, \tau_\alpha, b_\alpha, \beta, \gamma_\beta, \tau_\beta, b_\beta, \mathbf{p}, \mathbf{q}, \mathbf{F}_{ST}, \mu_{F_{ST}}, \mathbf{F}, \mu_F, \gamma_F, \varepsilon)$ is given by Bayes' rule:

$$f(\mathbf{\Theta}|\mathbf{G}, \mathbf{S}, \mathbf{D}) \propto \Pr(\mathbf{G}|\mathbf{S}, \alpha, \beta, \mathbf{p}, \mathbf{F}, \varepsilon) f(\alpha|\gamma_\alpha, \tau_\alpha, b_\alpha) f(\gamma_\alpha) f(\tau_\alpha) f(b_\alpha|\mathbf{D})$$
$$f(\beta|\gamma_\beta, \tau_\beta, b_\beta) f(\gamma_\beta) f(\tau_\beta) f(b_\beta|\mathbf{D}) \times f(\mathbf{p}|\mathbf{q}, \mathbf{F}_{ST}) f(\mathbf{q}) f(\mathbf{F}_{ST}|\mu_{F_{ST}}) \quad (8)$$
$$f(\mu_{F_{ST}}) f(\mathbf{F}|\mu_F, \gamma_F) f(\mu_F) f(\gamma_F) f(\varepsilon)$$

where $f$ on the right hand of the equation denotes a prior distribution of parameters, as defined above. We estimated the joint posterior distribution of parameters using a data augmentation strategy (Tanner and Wong 1987), treating as latent variables the population origin of individuals $\mathbf{o}$ and additional indicator variables. This approach is well suited to mixture models, simplifying the estimation algorithm substantially because many of the parameters can be estimated by direct sampling from the conditional posterior distribution. Details of the used MCMC algorithms are described in Supplement B in Supplementary File 1.

## Simulation study of method performance

We used Monte Carlo simulations to assess the bias and accuracy (root mean square error, RMSE) of $\mathbf{\Theta}$ posterior estimates, as well as the ability of the method to infer distance effects on seed and pollen migration. We investigated the method performance under different simulated sample sizes, population numbers, genotypic assays, levels of population genetic divergence, individual inbreeding values, and degrees of isolation by distance in seed and pollen migration. Due to computational resource limitations, we could not conduct a broad and intensively replicated exploration of parameter space, so we focused on a representative set of simulated scenarios with sufficient replication to reveal the sensitivity of estimates to major factors. The parameterization of the stochastic algorithm used for the simulations followed the inference model, as detailed in Supplement C in Supplementary File 1.

We considered a first set of scenarios in which data were simulated without distance effects on migration and where distance effects were assumed to be null during inference. For this initial set, we simulated $K = 10$ populations and randomly draw seed and pollen migrants into any given population from any of the other $K - 1$ populations, i.e. we assumed $b_\alpha = b_\beta = 0$. The values assumed for the rest of the migration parameters were as follows: prior probabilities of local dispersal $\lambda_\alpha = \lambda_\beta = 2/3$, isolation parameters $\tau_\alpha = \tau_\beta = 0.25$, and dispersion parameters $\gamma_\alpha = \gamma_\beta = 0.1$. These migration parameter values correspond to a mean expected pairwise migration rate of $\alpha_{ij} = \beta_{ij} = 0.25/9 = 0.028$, equal for seed and pollen, with realized pairwise migration rates ranging stochastically in the simulations (independently for seed and pollen) between 0 and 0.5. We evaluated individual sample size effects by considering three sampling intensities, $N = \{1000, 500, 250\}$. As a benchmark for subsequent scenarios, we first assessed the method performance with sampling effects but under ideal discrimination conditions, i.e. assuming that the origin of seed and pollen migrants could be categorically determined, and therefore neglecting the model's genetic parameters during simulation and inference. Each of the three reference ideal scenarios was replicated 100 times. We then simulated more realistic scenarios in which population origin was inferred based on genetic marker data, considering the same three $N$ values while also assessing the influence on parameter estimation of population genetic differentiation, individual inbreeding, and population number, with assumed values of $\mu_{F_{ST}} = \{0.2, 0.1, 0.05, 0.025\}$, $\mu_F = \{0, 0.1, 0.2\}$, and $K = \{10, 20\}$, respectively. The effect of increasing population number from $K = 10$ to 20 was considered in a restricted number of scenarios,

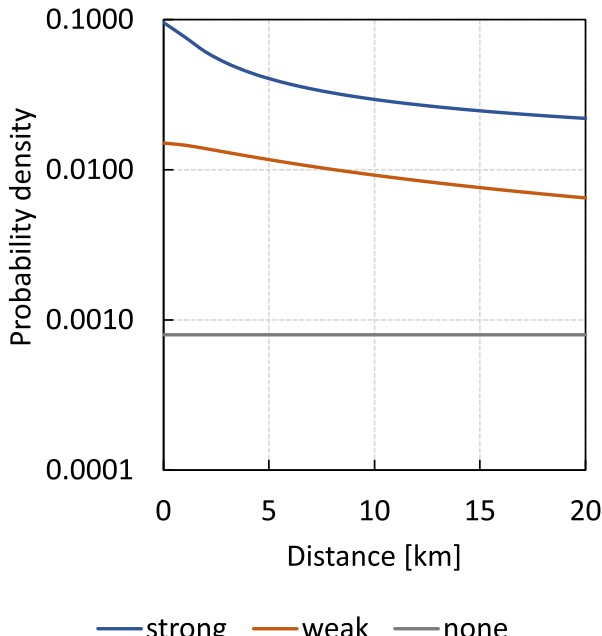

**Fig. 1.** Probability density functions of migration distances from a population source (dispersal kernels) assumed in the simulations. Note the logarithmic scale on the y-axis. Each kernel was obtained by setting a different distance effect parameter (*b*) value, in order to simulate contrasting isolation by distance patterns: strong (*b* = 2.1274), weak (*b* = 1.2062), and null (*b* = 0). The probability of migration beyond 20 km was assumed negligible (see Supplement C in Supplementary File 1).

namely with $\mu_{F_{ST}} = 0.1$ and $\mu_F = 0$. The mean expected pairwise migration rate of the scenarios with $K = 20$ was $\alpha_{ij} = \beta_{ij} = 0.25/19 = 0.013$. In all cases, we simulated two alternative genotyping assays, an SSR-type one, consisting of $L = 20$ loci with 6 alleles each, and an SNP-type one, with $L = 1,000$ loci with 2 alleles each. There were 78 different scenarios with genetic inference of migrant origin, each one replicated 10 times, yielding a total of 780 replicates. We used the simulated data without distance effects to assess the bias and RMSE of estimates of seed and pollen migration ($\alpha_{ij}$ and $\beta_{ij}$), population genetic divergence ($F_{ST}^j$), and individual inbreeding ($F_i$).

In a second set of scenarios, we considered distance effects on migration during simulations and inference. The assumed spatial distribution of the simulated populations followed a real one, namely the distribution of 9 European yew (*Taxus baccata* L.) isolated remnant populations in Western Carpathians (Poland), with pairwise separation distances ranging from 0.84 to 19.64 km (Chybicki *et al.* 2024; see case study below). We thus considered $K = 9$ populations with this spatial distribution and conducted simulations as described above, except that pairwise migration rates now decreased with the distance between populations. Specifically, we assumed 3 different levels of isolation by distance (strong, weak, and null) by setting $b_\alpha$ and $b_\beta$ values independently at 2.1274, 1.2062, or 0, respectively. The resulting median migration distances were 5, 10, and 14.14 km, respectively (Fig. 1). We assumed the same range of values for $N$ and $\mu_{F_{ST}}$ as in the simulations without distance effects, considering now for simplicity inbreeding absence and SSR-type genetic markers only. We evaluated 9 different combinations of isolation by distance in seed and pollen migration, 3 sampling efforts, and 5 population divergence levels. In addition, setting $\mu_{F_{ST}}$ at 0.1 and seed and pollen isolation by distance at an intermediate level

($b_\alpha = b_\beta = 1.2062$), we evaluated the effect of a larger number of populations within the study region, namely $K = 18$ vs 9. We generated the spatial coordinates of the 9 additional populations as opposites of the original 9, considering their barycenter as coordinate origin. Overall, there were 111 different scenarios with distance effects, totaling 1,110 replicates (10 replicates/scenario). In these scenarios, we focused on the bias and RMSE of estimates of seed and pollen migration ($\alpha_{ij}$ and $\beta_{ij}$) and distance effect ($b_\alpha$ and $b_\beta$). Besides the accuracy in $b_\alpha$ and $b_\beta$ estimation, we also assessed the model ability to identify correctly the presence or absence of distance effects (as described in Supplement C in Supplementary File 1).

### Real data example

We applied our model to available real genotypic data from a geographically isolated network of nine remnant populations of the gymnosperm tree *T. baccata* (Chybicki 2024), distributed across the Low Beskids (Western Carpathians, Poland) with the above mentioned range of interpopulation pairwise distances (between 0.84 and 19.64 km). In a previous study (Chybicki *et al.* 2024), a total of 1,167 adult trees were sampled from the nine populations (mean 130 trees/population, range 42–434) and genotyped at 20 SSR loci to estimate seed and pollen migration rates among naturally regenerated seedling samples, using the ESPM model (Robledo-Arnuncio and Gaggiotti 2017). In the previous study, AMOVA results showed substantial genetic divergence among the yew remnants ($F_{ST} = 0.13$), while estimated migration rates among the seedling samples were barely detectable for seeds and larger (up to 1.1%) and significant for pollen, showing a decreasing trend with geographical distance in an independent correlation analysis. In the present study, we use the same adult genotypic samples (but not the seedling ones) and apply our new inference model to estimate jointly seed and pollen migration rates during the last generation (instead of among contemporary recruits) and distance effects on migration.

We assumed the full inference model (4), considering both **F** and $\varepsilon$ as nonzero parameters to be estimated. The posterior distribution of parameters was obtained using the MCMC algorithm described in Supplement B in Supplementary File 1, with 10,000 burn-in cycles followed by 20,000 cycles thinned to every 10th, yielding a final sample of 2,000.

### Results
#### Inference without distance effects

We consider the model without distance effects, i.e. where the assumed proportion of migrants are independently randomly drawn from any of the $K - 1$ external populations to generate simulated data sets, and distance effects are assumed to be null during inference (i.e. $b_\alpha$ and $b_\beta$ are fixed at zero). We did not observe convergence problems in MCMC runs for any of the corresponding simulated data sets. The reference ideal scenarios with categorical discrimination of migrant origin showed virtually null biases in seed ($\alpha_{ij}$) and pollen ($\beta_{ij}$) pairwise migration rate estimates, while the sampling variance resulted in RMSEs ranging from a low of around 0.015 (i.e. $0.015/0.028 \approx 54\%$ in relative terms) for a sample size of $N = 1,000$ to a high of around 0.027 ($\sim99\%$) for $N = 250$, similarly for both dispersal vectors (Table 1 and Fig. 2). In the more realistic scenarios where migrant origin has to be inferred probabilistically based on observed genotypes, migration estimation errors increased with decreasing population genetic divergence ($\mu_{F_{ST}}$) and decreasing $N$, more so for pollen than for seeds, as described below.

**Table 1.** Effect of mean population genetic differentiation ($\mu_{F_{ST}}$) and total sample size ($N$) on the bias and RMSE of estimates of migration rates, population divergence and inbreeding, assuming microsatellite-type markers, no inbreeding ($\mu_F = 0$), and no distance effect on migration rates.

| $\mu_{F_{ST}}$ | $N$ | Seed migration ($\alpha_{ij}$) | | Pollen migration ($\beta_{ij}$) | | Population divergence ($F^j_{ST}$) | | Individual inbreeding ($F_i$) | |
|---|---|---|---|---|---|---|---|---|---|
| | | Bias | RMSE | Bias | RMSE | Bias | RMSE | Bias | RMSE |
| Ideal[a] | 1000 | 0.0001 | 0.0147 | 0.0001 | 0.0154 | — | — | — | — |
| | 500 | 0.0000 | 0.0203 | 0.0000 | 0.0211 | — | — | — | — |
| | 250 | −0.0001 | 0.0275 | 0.0004 | 0.0268 | — | — | — | — |
| 0.200 | 1000 | 0.0007 | 0.0163 | −0.0003 | 0.0156 | −0.0092 | 0.0227 | 0.0033 | 0.0072 |
| | 500 | −0.0007 | 0.0235 | −0.0002 | 0.0230 | −0.0087 | 0.0216 | 0.0042 | 0.0088 |
| | 250 | 0.0012 | 0.0250 | 0.0005 | 0.0279 | −0.0068 | 0.0300 | 0.0054 | 0.0097 |
| 0.100 | 1000 | −0.0003 | 0.0156 | 0.0000 | 0.0221 | −0.0043 | 0.0140 | 0.0019 | 0.0036 |
| | 500 | −0.0004 | 0.0220 | −0.0007 | 0.0294 | −0.0012 | 0.0149 | 0.0027 | 0.0072 |
| | 250 | 0.0000 | 0.0392 | −0.0011 | 0.0430 | −0.0046 | 0.0195 | 0.0039 | 0.0086 |
| 0.050 | 1000 | 0.0009 | 0.0187 | −0.0028 | 0.0355 | −0.0018 | 0.0077 | 0.0013 | 0.0034 |
| | 500 | 0.0012 | 0.0258 | −0.0038 | 0.0472 | −0.0015 | 0.0119 | 0.0022 | 0.0059 |
| | 250 | −0.0017 | 0.0409 | −0.0030 | 0.0538 | −0.0018 | 0.0133 | 0.0022 | 0.0040 |
| 0.025 | 1000 | −0.0008 | 0.0314 | −0.0031 | 0.0462 | 0.0027 | −0.0011 | 0.0462 | 0.0012 |
| | 500 | −0.0024 | 0.0381 | −0.0014 | 0.0503 | 0.0052 | −0.0006 | 0.0503 | 0.0017 |
| | 250 | −0.0026 | 0.0497 | −0.0052 | 0.0568 | 0.0051 | −0.0040 | 0.0568 | 0.0025 |

Based on 10 independent replicates per scenario, assuming $L = 20$ loci, 6 alleles/locus, $K = 10$ populations, isolation parameters $\tau_\alpha = \tau_\beta = 0.25$, and dispersion parameters $\gamma_\alpha = \gamma_\beta = \gamma_F = 0.1$ and $\gamma_{F_{ST}} = 0.01$.
[a]Ideal reference scenario assuming categorical discrimination of migrant origins.

Considering first microsatellite-type markers (20 loci with 6 alleles each) and no inbreeding ($F_i = 0$), the bias of migration estimates was generally low, remaining below 5% in relative terms for $\mu_{F_{ST}} \geq 0.05$ or $\mu_{F_{ST}} \geq 0.10$ for $\alpha_{ij}$ and $\beta_{ij}$, respectively, independently of the considered sample sizes (Table 1). The biases increased to maximum relative values of around 9 and 19% for $\alpha_{ij}$ and $\beta_{ij}$, respectively, for the weakest population genetic structure ($\mu_{F_{ST}} = 0.025$) and smallest sample size ($N = 250$) considered (Table 1). Under strong population genetic divergence ($\mu_{F_{ST}} = 0.20$), the RMSE of migration estimates remained close to the corresponding reference minima attained under the ideal scenario of categorical origin determination (54–99%), but they progressively augmented with decreasing $\mu_{F_{ST}}$, up to an approximate two fold increase for the most unfavorable scenario with $\mu_{F_{ST}} = 0.025$ and $N = 250$ (Table 1 and Fig. 2). Along with seed and pollen migration estimates, the model jointly inferred population divergence ($F^j_{ST}$) and individual inbreeding ($F_i$) values rather accurately in general, with RMSE that remained low in all scenarios considered and biases that only increased notably for $F_i$ when $\mu_{F_{ST}} = 0.025$. On the other hand, the presence of individual inbreeding ($\mu_F = 0.1$ or $0.2$) had little effect on migration estimates, which showed rather similar bias and RMSE than without inbreeding across all sampling and population differentiation scenarios considered (Supplementary Tables S1 and S2 in Supplementary File 1 vs Table 1). Similarly, increasing individual inbreeding values had limited impact on $F^j_{ST}$ estimation errors, while they tended to increase the bias and RMSE of inbreeding estimates themselves (Supplementary Tables S1 and S2 in Supplementary File 1 vs Table 1).

Seed and pollen migration estimates improved when assuming 1,000 biallelic loci, rather than the microsatellite-type markers (Table 2 vs Table 1 and Fig. 3 vs Fig. 2). Specifically, using the SNP-type genotyping assay, the estimation bias of both $\alpha_{ij}$ and $\beta_{ij}$ remained now low (<5%) in all population divergence and sampling scenarios considered, even for the most unfavorable combination of $\mu_{F_{ST}} = 0.025$ and $N = 250$ (Table 2 and Fig. 3). And, notably, the RMSE of seed migration estimates remained close to the reference minimum values achieved under categorical discrimination of migrant origins in all cases except for $\mu_{F_{ST}} = 0.025$

combined with $N = 250$, so did the RMSE of pollen migration estimates, except for $\mu_{F_{ST}} = 0.05$ combined with $N = 250$ or when $\mu_{F_{ST}} = 0.025$ (Table 2). Estimates of $F^j_{ST}$ and $F_i$ were also more accurate for the SNP-type than for the microsatellite-type assay, with even lower bias and RMSE across all scenarios considered (Table 2 vs Table 1). Estimates of migration using the SNP-type markers were also weakly sensitive to increasing individual inbreeding (Supplementary Tables S3 and S4 in Supplementary File 1 vs Table 2).

For a given total sample size ($N$), increasing the number of simulated populations from $K = 10$ to $20$ did not have a strong impact on migration rate estimates (Supplementary Table S5 in Supplementary File 1). In particular, the absolute biases of $\alpha_{ij}$ and $\beta_{ij}$ remained low when doubling $K$, while their absolute RMSE even decreased in some cases, notably for the smallest considered sample size of $N = 250$ (Supplementary Table S5 in Supplementary File 1). The RMSE of $F^j_{ST}$ estimates consistently increased with larger $K$ for all the considered sample sizes for SSR-type markers, whereas it remained more stable when using bi-allelic (SNP-type) markers (Supplementary Table S5 in Supplementary File 1). By contrast, estimates of individual inbreeding were rather robust to $K$ variation across sample sizes for both marker types (Supplementary Table S5 in Supplementary File 1).

## Inference with distance effects

We now consider results of the simulated scenarios in which interpopulation distance has varying effects (null, weak, or strong) on migration rates, independently for seeds and pollen, jointly estimating these effects during the inference stage through $b_\alpha$ and $b_\beta$. The increase in parameter dimensionality did not generate MCMC convergence problems for any of the runs of any of the simulated scenarios, nor did the joint estimation of distance effects have a strong influence on the accuracy of seed and pollen migration rate estimates. In particular, the bias and RMSE of $\alpha_{ij}$ and $\beta_{ij}$ with joint estimation of distance effects remained around (slightly above or below, depending on the scenario) the values attained in simulations in which distance effects did not exist and were not estimated, similarly increasing under

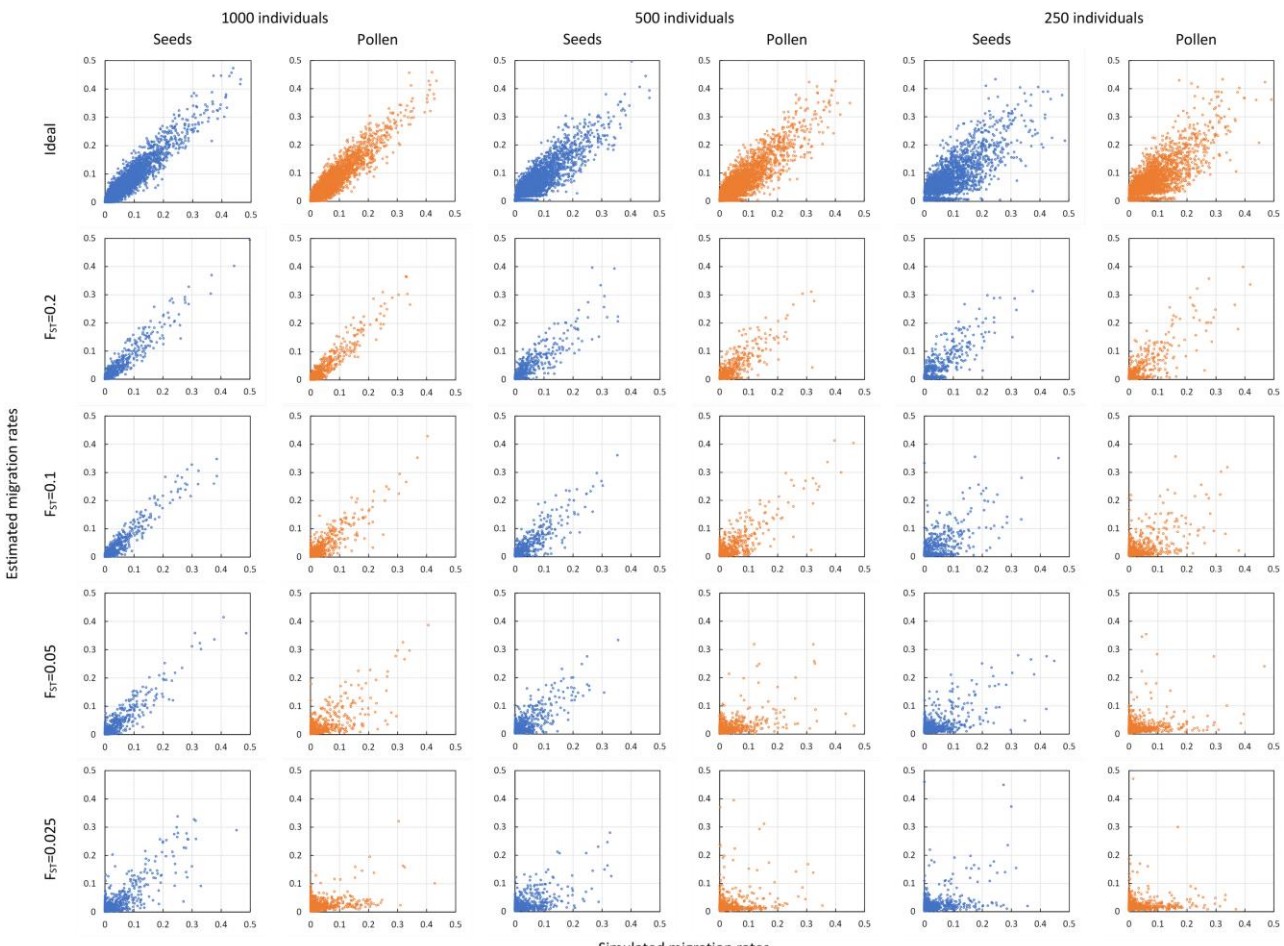

**Fig. 2.** Estimated vs simulated seed and pollen migration rates assuming SSR-type markers. Panel rows illustrate the effect on migration estimates of genetic discrimination power: the top row shows benchmark scenarios with ideal categorical discrimination, while the following four assume, in descending order, $F_{ST} = 0.20$, 0.10, 0.05, and 0.025, respectively. Panel columns show the effect of decreasing total sample size, in pairs from left to right: $N = 1,000$ (columns 1 and 2), $N = 500$ (columns 3 and 4), and $N = 250$ (columns 5 and 6). Based on 10 (or 100 in scenarios with categorical discrimination) independent simulations replicates per scenario, assuming $L = 20$ loci, 6 alleles/locus, $K = 10$ populations, no inbreeding ($\mu_F = 0$), no distance effect on migration, isolation parameters $\tau_\alpha = \tau_\beta = 0.25$, and dispersion parameters $\gamma_\alpha = \gamma_\beta = \gamma_F = 0.1$ and $\gamma_{F_{ST}} = 0.01$.

**Table 2.** Effect of mean population genetic differentiation ($\mu_{F_{ST}}$) and total sample size (N) on the bias and RMSE of estimates of migration rates, population divergence and inbreeding, assuming SNP-type markers, no inbreeding ($\mu_F = 0$), and no distance effect on migration rates.

| $\mu_{F_{ST}}$ | N | Seed migration ($\alpha_{ij}$) | | Pollen migration ($\beta_{ij}$) | | Population divergence ($F_{ST}^j$) | | Individual inbreeding ($F_i$) | |
|---|---|---|---|---|---|---|---|---|---|
| | | Bias | RMSE | Bias | RMSE | Bias | RMSE | Bias | RMSE |
| Ideal[a] | 1000 | 0.0001 | 0.0147 | 0.0001 | 0.0154 | — | — | — | — |
| | 500 | 0.0000 | 0.0203 | 0.0000 | 0.0211 | — | — | — | — |
| | 250 | −0.0001 | 0.0275 | 0.0004 | 0.0268 | — | — | — | — |
| 0.200 | 1000 | 0.0000 | 0.0150 | −0.0004 | 0.0167 | −0.0062 | 0.0102 | 0.0010 | 0.0031 |
| | 500 | 0.0004 | 0.0200 | 0.0003 | 0.0220 | −0.0062 | 0.0093 | 0.0009 | 0.0017 |
| | 250 | 0.0006 | 0.0283 | 0.0005 | 0.0285 | −0.0074 | 0.0111 | 0.0012 | 0.0021 |
| 0.100 | 1000 | 0.0004 | 0.0151 | −0.0006 | 0.0157 | −0.0014 | 0.0045 | 0.0010 | 0.0018 |
| | 500 | −0.0001 | 0.0217 | 0.0003 | 0.0195 | −0.0017 | 0.0048 | 0.0009 | 0.0022 |
| | 250 | −0.0013 | 0.0303 | 0.0000 | 0.0274 | −0.0017 | 0.0060 | 0.0014 | 0.0028 |
| 0.050 | 1000 | 0.0009 | 0.0162 | 0.0004 | 0.0167 | −0.0002 | 0.0024 | 0.0007 | 0.0016 |
| | 500 | −0.0009 | 0.0219 | 0.0010 | 0.0220 | −0.0005 | 0.0034 | 0.0008 | 0.0022 |
| | 250 | −0.0015 | 0.0276 | 0.0006 | 0.0300 | −0.0008 | 0.0036 | 0.0011 | 0.0025 |
| 0.025 | 1000 | −0.0002 | 0.0145 | −0.0002 | 0.0184 | 0.0000 | 0.0017 | 0.0007 | 0.0019 |
| | 500 | 0.0006 | 0.0230 | 0.0008 | 0.0254 | −0.0001 | 0.0020 | 0.0007 | 0.0021 |
| | 250 | −0.0007 | 0.0318 | −0.0007 | 0.0378 | −0.0007 | 0.0037 | 0.0006 | 0.0015 |

Based on 10 independent replicates per scenario, assuming $L = 1,000$ loci, 2 alleles/locus, $K = 10$ populations, isolation parameters $\tau_\alpha = \tau_\beta = 0.25$, and dispersion parameters $\gamma_\alpha = \gamma_\beta = \gamma_F = 0.1$ and $\gamma_{F_{ST}} = 0.01$.
[a]Ideal reference scenario assuming categorical discrimination of migrant origins.

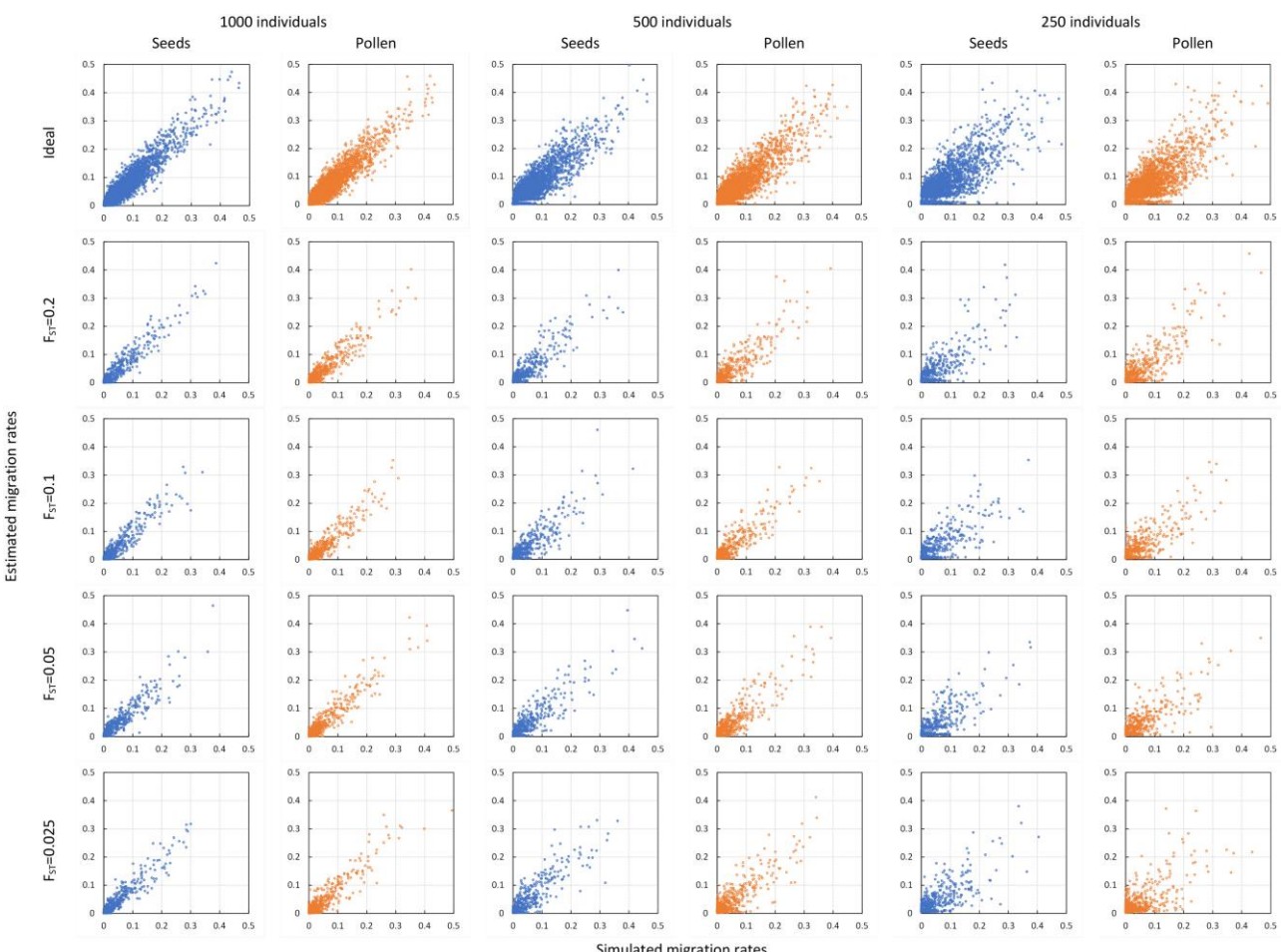

**Fig. 3.** Estimated vs simulated seed and pollen migration rates assuming SNP-type markers. Panel rows illustrate the effect on migration estimates of genetic discrimination power: the top row shows benchmark scenarios with ideal categorical discrimination, while the following 4 assume, in descending order, $F_{ST} = 0.20$, 0.10, 0.05, and 0.025, respectively. Panel columns show the effect of decreasing total sample size, in pairs from left to right: $N = 1,000$ (columns 1 and 2), $N = 500$ (columns 3 and 4), and $N = 250$ (columns 5 and 6). Based on 10 (or 100 in scenarios with categorical discrimination) independent simulations replicates per scenario, assuming $L = 1,000$ loci, 2 alleles/locus, $K = 10$ populations, no inbreeding ($\mu_F = 0$), no distance effect on migration, isolation parameters $\tau_\alpha = \tau_\beta = 0.25$, and dispersion parameters $\gamma_\alpha = \gamma_\beta = \gamma_F = 0.1$ and $\gamma_{F_{ST}} = 0.01$.

weaker population genetic structures and smaller sample sizes (Table 3; Supplementary Tables S6 and S7 in Supplementary File 1 vs Table 1). Moreover, the assumed magnitude of distance effects did not have a clear impact on migration rate estimation, because even if the bias and RMSE of $\alpha_{ij}$ and $\beta_{ij}$ exhibited some variation among the different assumed combinations of $b_\alpha$ and $b_\beta$ values, this variation was not large and might have been at least partly due to variance among stochastic replicates, as it was not consistent across different genetic divergence and sample size scenarios (Table 3; Supplementary Tables S6 and S7 in Supplementary File 1).

Regarding inference of distance effects themselves, the model successfully detected them when they were present, or discarded them when they were absent, adequately discriminating between independent effects on seed vs pollen migration, provided sufficient sample size and genetic divergence among populations (Table 3 and Supplementary Tables S6 and S7 in Supplementary File 1). In particular, considering first simulated scenarios with a decreasing probability of seed migration with distance ($b_\alpha > 0$) and moderate-to-strong genetic divergence ($\mu_{F_{ST}} \geq 0.1$), the reversible-jump MCMC algorithm allowed correctly selecting the model including $b_\alpha$ as the best model in 100% (i.e. 10 out of 10) of the independent replicates for $N = 1,000$ (Table 3), 90–100% for $N = 500$ (Supplementary Table S6 in Supplementary File 1), and

80–100% for $N = 250$ (Supplementary Table S7 in Supplementary File 1), with lower percentages corresponding to the cases where assumed distance effects were weak ($b_\alpha = 1.2062$) rather than strong ($b_\alpha = 2.1274$). When the assumed level of genetic divergence decreased to $\mu_{F_{ST}} = 0.05$, the model including $b_\alpha$ was still correctly identified as the best model 90–100% of times for $N \geq 500$ (Table 3; Supplementary Table S6 in Supplementary File 1), but only 30–80% of times if the sample size was small ($N = 250$) and the distance effect on seed migration was weak (Supplementary Table S7 in Supplementary File 1). Under the lowest assumed genetic divergence value, $\mu_{F_{ST}} = 0.025$, the power to detect $b_\alpha > 0$ was still as high as 70–100% for $N = 1,000$ and also for $N = 500$ when the distance effect on seed migration was strong, decreasing to 30–80% otherwise (Table 3; Supplementary Tables S6 and S7 in Supplementary File 1). The power of the model to detect decreasing pollen migration rates with distance ($b_\beta > 0$) was nearly as high as to detect seed migration effects when population genetic divergence was high ($\mu_{F_{ST}} \geq 0.1$), but it became comparatively lower for $\mu_{F_{ST}} \leq 0.05$, down to a power of 10–30% for the most unfavorable combination of $\mu_{F_{ST}} = 0.025$ and $N = 250$ (Table 3; Supplementary Tables S6 and S7 in Supplementary File 1).

Based on the simulated scenarios with spatially uniform seed migration ($b_\alpha = 0$), the type I error rate (estimated as the

**Table 3.** Effect of mean population genetic differentiation ($\mu_{F_{ST}}$) and magnitude of interpopulation distance effects on seed and pollen migration ($b_\alpha$ and $b_\beta$, respectively) on the bias and RMSE of estimates of the distance effects and of seed and pollen migration rates ($\alpha_{ij}$ and $\beta_{ij}$, respectively), assuming a total sample size of $N = 1,000$ individuals. The model ability to identify correctly the presence or absence of distance effects was characterized by the number of independent simulation replicates ($n$ pos) in which the reversible-jump MCMC algorithm selected the model including $b_\alpha$ (or $b_\beta$) as the best model, which was based on the proportion of times (Pr) it was visited during the sampled MCMC cycles.

| | | | Seed migration | | | | | | Pollen migration | | | | | |
|---|---|---|---|---|---|---|---|---|---|---|---|---|---|---|
| $\mu_{F_{ST}}$ | $b_\alpha$ | $b_\beta$ | Bias ($\alpha_{ij}$) | RMSE ($\alpha_{ij}$) | Bias ($b_\alpha$) | RMSE ($b_\alpha$) | Pr($b_\alpha$) | $n$ pos | Bias ($\beta_{ij}$) | RMSE ($\beta_{ij}$) | Bias ($b_\beta$) | RMSE ($b_\beta$) | Pr($b_\beta$) | $n$ pos |
| 0.200 | 2.1274 | 2.1274 | 0.0001 | 0.0159 | 0.0814 | 0.3018 | 0.9967 | 10 | −0.0004 | 0.0178 | 0.0043 | 0.2755 | 0.9962 | 10 |
| | | 1.2062 | 0.0002 | 0.0171 | −0.0093 | 0.2363 | 0.9978 | 10 | −0.0006 | 0.0178 | 0.0882 | 0.4668 | 0.9970 | 10 |
| | | 0 | −0.0002 | 0.0156 | 0.0785 | 0.4329 | 0.9965 | 10 | 0.0004 | 0.0153 | −0.0844 | 0.2668 | 0.1925 | 1 |
| | 1.2062 | 2.1274 | −0.0004 | 0.0171 | 0.1830 | 0.3688 | 0.9970 | 10 | 0.0000 | 0.0167 | −0.0372 | 0.3797 | 0.9979 | 10 |
| | | 1.2062 | 0.0001 | 0.0160 | 0.0088 | 0.2892 | 0.9436 | 10 | 0.0002 | 0.0175 | −0.0521 | 0.2347 | 0.9914 | 10 |
| | | 0 | −0.0001 | 0.0163 | −0.0404 | 0.2746 | 0.9854 | 10 | 0.0006 | 0.0166 | 0.0000 | 0.0000 | 0.1164 | 0 |
| | 0 | 2.1274 | −0.0008 | 0.0171 | 0.0000 | 0.0000 | 0.0692 | 0 | 0.0003 | 0.0163 | −0.1448 | 0.3662 | 0.9976 | 10 |
| | | 1.2062 | −0.0006 | 0.0142 | 0.0000 | 0.0000 | 0.0636 | 0 | 0.0004 | 0.0151 | 0.0695 | 0.4860 | 0.9217 | 9 |
| | | 0 | 0.0001 | 0.0155 | −0.2496 | 0.4596 | 0.2820 | 3 | 0.0004 | 0.0172 | −0.0728 | 0.2301 | 0.1309 | 1 |
| 0.100 | 2.1274 | 2.1274 | −0.0001 | 0.0154 | 0.1727 | 0.2326 | 0.9953 | 10 | −0.0006 | 0.0201 | −0.0965 | 0.3439 | 0.9946 | 10 |
| | | 1.2062 | 0.0009 | 0.0175 | −0.0449 | 0.2909 | 0.9976 | 10 | −0.0008 | 0.0212 | −0.0776 | 0.4230 | 0.9297 | 9 |
| | | 0 | 0.0009 | 0.0145 | −0.1144 | 0.3093 | 0.9979 | 10 | −0.0001 | 0.0207 | 0.0000 | 0.0000 | 0.0404 | 0 |
| | 1.2062 | 2.1274 | 0.0005 | 0.0151 | −0.0367 | 0.3708 | 0.9440 | 10 | 0.0010 | 0.0219 | 0.0042 | 0.1706 | 0.9931 | 10 |
| | | 1.2062 | −0.0003 | 0.0170 | 0.0882 | 0.3737 | 0.9752 | 10 | 0.0006 | 0.0218 | 0.0514 | 0.5221 | 0.8999 | 9 |
| | | 0 | −0.0001 | 0.0157 | 0.0558 | 0.3551 | 0.9737 | 10 | 0.0009 | 0.0216 | 0.0000 | 0.0000 | 0.1313 | 0 |
| | 0 | 2.1274 | 0.0008 | 0.0176 | 0.0000 | 0.0000 | 0.0878 | 0 | 0.0001 | 0.0182 | −0.0045 | 0.3075 | 0.9894 | 10 |
| | | 1.2062 | 0.0001 | 0.0159 | 0.0000 | 0.0000 | 0.0345 | 0 | 0.0006 | 0.0208 | 0.0787 | 0.3322 | 0.9370 | 10 |
| | | 0 | 0.0005 | 0.0159 | 0.0000 | 0.0000 | 0.0799 | 0 | −0.0007 | 0.0243 | 0.0000 | 0.0000 | 0.0896 | 0 |
| 0.050 | 2.1274 | 2.1274 | −0.0007 | 0.0195 | 0.0364 | 0.3584 | 0.9957 | 10 | 0.0003 | 0.0306 | −0.3703 | 0.8344 | 0.9221 | 9 |
| | | 1.2062 | −0.0001 | 0.0204 | −0.0513 | 0.3610 | 0.9943 | 10 | −0.0028 | 0.0330 | −0.3654 | 0.8177 | 0.5529 | 6 |
| | | 0 | 0.0009 | 0.0169 | −0.1827 | 0.3081 | 0.9953 | 10 | −0.0020 | 0.0360 | 0.0000 | 0.0000 | 0.0565 | 0 |
| | 1.2062 | 2.1274 | −0.0009 | 0.0200 | 0.0314 | 0.3120 | 0.9350 | 10 | −0.0012 | 0.0358 | −0.3766 | 0.4626 | 0.9235 | 10 |
| | | 1.2062 | −0.0004 | 0.0238 | −0.1637 | 0.4828 | 0.8427 | 9 | −0.0026 | 0.0433 | 0.3301 | 0.7119 | 0.8580 | 9 |
| | | 0 | −0.0005 | 0.0225 | −0.0313 | 0.5100 | 0.8756 | 9 | −0.0013 | 0.0394 | 0.0000 | 0.0000 | 0.0800 | 0 |
| | 0 | 2.1274 | −0.0003 | 0.0203 | 0.0000 | 0.0000 | 0.0641 | 0 | −0.0011 | 0.0376 | −0.1843 | 0.3729 | 0.9731 | 10 |
| | | 1.2062 | 0.0011 | 0.0199 | 0.0731 | 0.2313 | 0.1370 | 1 | −0.0031 | 0.0366 | 0.0511 | 0.8362 | 0.6816 | 7 |
| | | 0 | 0.0009 | 0.0205 | 0.0000 | 0.0000 | 0.0325 | 0 | −0.0039 | 0.0390 | 0.1165 | 0.5999 | 0.3043 | 3 |
| 0.025 | 2.1274 | 2.1274 | −0.0010 | 0.0283 | 0.1678 | 0.4156 | 0.9497 | 10 | −0.0051 | 0.0540 | −1.1285 | 1.5223 | 0.5369 | 5 |
| | | 1.2062 | 0.0004 | 0.0251 | −0.1916 | 0.4120 | 0.9882 | 10 | −0.0030 | 0.0494 | −0.8063 | 1.1512 | 0.3762 | 2 |
| | | 0 | 0.0010 | 0.0258 | −0.0644 | 0.2160 | 0.9946 | 10 | −0.0072 | 0.0517 | 0.0000 | 0.0000 | 0.1521 | 0 |
| | 1.2062 | 2.1274 | −0.0002 | 0.0292 | 0.3451 | 0.5310 | 0.9544 | 10 | −0.0070 | 0.0564 | −1.7915 | 1.9133 | 0.3254 | 2 |
| | | 1.2062 | −0.0014 | 0.0264 | −0.0805 | 0.6665 | 0.7697 | 8 | −0.0021 | 0.0487 | −0.6975 | 1.0502 | 0.4090 | 3 |
| | | 0 | −0.0001 | 0.0280 | −0.1943 | 0.7429 | 0.7752 | 7 | −0.0058 | 0.0528 | 0.0000 | 0.0000 | 0.1253 | 0 |
| | 0 | 2.1274 | −0.0012 | 0.0285 | −0.1264 | 0.3997 | 0.1175 | 1 | −0.0022 | 0.0543 | −1.0454 | 1.4048 | 0.5557 | 6 |
| | | 1.2062 | −0.0008 | 0.0297 | 0.0000 | 0.0000 | 0.1052 | 0 | −0.0061 | 0.0523 | −0.9138 | 1.0907 | 0.3136 | 2 |
| | | 0 | 0.0012 | 0.0303 | 0.0000 | 0.0000 | 0.1060 | 0 | −0.0055 | 0.0561 | 0.0000 | 0.0000 | 0.1636 | 0 |

Based on 10 independent replicates per scenario, assuming $L = 20$ loci with 6 alleles/locus, $K = 9$ populations, no inbreeding ($\mu_F = 0$), isolation parameters $\tau_\alpha = \tau_\beta = 0.25$, and dispersion parameters $\gamma_\alpha = \gamma_\beta = \gamma_F = 0.1$.

proportion of independent replicates in which the algorithm incorrectly selected the model with $b_\alpha > 0$ as the best model) was generally low, being 0% in 28 out of the 36 corresponding scenarios, 10% in 6 scenarios, and 30% in 2 scenarios (yielding an average of 3.3% across all scenarios; Table 3; Supplementary Tables S6 and S7 in Supplementary File 1). No evident association was observed between the type I error rate for $b_\alpha$ and sample size, population genetic divergence, or the independent distance effect on pollen migration ($b_\beta$). As for the scenarios with spatially uniform pollen migration ($b_\beta = 0$), the estimated type I error rate was similarly low than for uniform seed migration, being 0% in 28 out of 36 scenarios, 10% in 4 scenarios, 20% in 2 scenarios, and 30% in 2 other scenarios (again without a clear relationship with sample size, population divergence or $b_\alpha$), translating into an overall mean error rate of 3.9% (Table 3; Supplementary Tables S6 and S7 in Supplementary File 1).

Regarding the point posterior estimates of $b_\alpha$ and $b_\beta$ parameters, simulation results indicated that their expected accuracy was dependent on population genetic divergence and sample size. Assuming $N = 1,000$, estimates of $b_\alpha$ showed generally low biases,

of around 5% relative to the assumed values (observed average across scenarios with $b_\alpha \neq 0$) for $\mu_{F_{ST}} \geq 0.05$ and 12% on average for $\mu_{F_{ST}} = 0.025$, with the average RMSE increasing from around 20 to 35% as $\mu_{F_{ST}}$ decreased from 0.2 to 0.025 (Table 3). The largest errors were observed for the smallest sample ($N = 250$), with the average relative bias of $b_\alpha$ ranging from 7 to 39%, and the average RMSE from 26 to 68%, for decreasing $\mu_{F_{ST}}$ from 0.2 to 0.025 (Supplementary Table S7 in Supplementary File 1), while the corresponding errors for $N = 500$ were intermediate between the ones observed for the largest and smallest sample sizes (Supplementary Table S6 in Supplementary File 1). The bias and RMSE of estimates of the distance effect on pollen migration ($b_\beta$) were around the same magnitude on average than those of $b_\alpha$ when $\mu_{F_{ST}} \geq 0.1$ and $N \geq 500$, but they became substantially larger when population differentiation was weak, sample size small or both simultaneously (Table 3; Supplementary Tables S6 and S7 in Supplementary File 1).

When distance effects are present, increasing the number of simulated populations from $K = 9$ to 18, while keeping total sample size ($N$) constant, tended to improve the estimation of both

**Table 4.** Posterior model parameter estimates for the *T. baccata* data set. The mean, standard error (SE), and 95% highest posterior density interval (HPDI) were calculated based on the posterior distribution of each parameter.

| Parameter | Description | Mean | SE | 95% HPDI |
|---|---|---|---|---|
| $\mu_{F_{ST}}$ | Mean population divergence | 0.2536 | 0.1044 | (0.1045, 0.4655) |
| $\mu_F$ | Mean individual inbreeding | 0.0072 | 0.0030 | (0.0012, 0.0125) |
| $\gamma_F$ | Dispersion of individual inbreeding | 0.2480 | 0.0981 | (0.0374, 0.4116) |
| $\gamma_\alpha$ | Dispersion of seed migration rates | 0.4279 | 0.1874 | (0.0614, 0.7452) |
| $\gamma_\beta$ | Dispersion of pollen migration rates | 0.0721 | 0.0483 | (0.0097, 0.1681) |
| $\tau_\alpha$ | Isolation due to seed nonmigration | 0.7078 | 0.1935 | (0.3110, 0.9874) |
| $\tau_\beta$ | Isolation due to pollen nonmigration | 0.7042 | 0.1071 | (0.4882, 0.8810) |
| $\Pr(b_\alpha > 0)$ | Probability of spatially dependent seed migration | 0.0864 | | |
| $\Pr(b_\beta > 0)$ | Probability of spatially dependent pollen migration | 0.9874 | | |
| $b_\alpha$ | Distance effect (scale) for seed migration | 0.5294 | 0.5299 | (−0.5646, 1.4930) |
| $b_\beta$ | Distance effect (scale) for pollen migration | 1.6199 | 0.5023 | (0.6608, 2.5959) |

migration rates and distance effect parameters themselves (Supplementary Table S8 in Supplementary File 1). The augmented pairwise spatial information that the model gains from larger *K* compensated the per-population data loss, consistently reducing the RMSE of estimates of $\alpha_{ij}$, $\beta_{ij}$, $b_\alpha$, and $b_\beta$ across all 3 considered sample sizes (Supplementary Table S8 in Supplementary File 1). On the other hand, the power of the model to detect distance effects on seed migration remained as high as 90–100% for all *N* when doubling *K*, while the power to detect distance effects on pollen migration remained high or even increased for $N \geq 500$ but decreased for the smallest sample size of $N = 250$ (Supplementary Table S8 in Supplementary File 1).

### Real data example

Posterior estimates of recent pairwise migration rates among the nine *T. baccata* populations ranged from very low minimum values ($<10^{-4}$) to a maximum of 7.9% for seeds and from similarly low values to a higher maximum of 11.6% for pollen (Supplementary Table S9 in Supplementary File 1). Most pairwise migration estimates were not significantly different from 0, as indicated by lower 95% HPDI limits below numerical precision ($<10^{-4}$), except for seed migration from Pa to Sr populations and for pollen migration from Sr to Pa (Supplementary Table S9 in Supplementary File 1). The posterior median estimates of seed migration rates were lower than those of pollen migration rates for all population pairs (Supplementary Table S9 in Supplementary File 1), which would suggest stronger demographic than reproductive isolation. However, such difference was not directly supported by estimates of the model isolation parameters for seed ($\tau_\alpha$) and pollen ($\tau_\beta$), which were very similar (Table 4). Variation in pairwise migration rates tended to be higher for seeds than for pollen, as revealed by the estimates of the dispersion parameters ($\gamma_\alpha > \gamma_\beta$; Table 4).

Seed migration rates were visually higher between neighboring yew populations (Fig. 4), but the distance effect for seed migration had little statistical support, with posterior probabilities of 0.09 and 0.91 for models with and without spatially dependent seed migration, respectively (Table 4). In the case of pollen, the negative association between migration and distance was clearer (Fig. 4) and had strong statistical support, with a posterior probability of 0.99 (Table 4). The posterior estimate of the slope of the spatial effect on pollen migration was close to the intermediate value used in the simulations to generate weak isolation by distance ($b_\beta = 1.62$, 95% HPDI 0.66–2.60; Table 4 and Fig. 1).

The posterior estimates indicate high genetic divergence before migration among the yew remnant populations ($\mu_{F_{ST}} = 0.254$), with

individual population values ranging from a low of 0.066 (Ly) to a high of 0.426 (Wa) (Supplementary Fig. S1 in Supplementary File 1). After accounting for allelic dropout, the studied yew individuals showed little inbreeding, with an estimated mean value of $\mu_F = 0.007$ with lower 95% HPDI limit of 0.001 (Table 4). Out of the 20 used loci, 4 showed posterior allelic dropout rate distributions not overlapping with 0 (Supplementary Fig. S2 in Supplementary File 1).

## Discussion

We have presented a new spatially explicit Bayesian model for the estimation of recent seed and pollen migration rates and the simultaneous assessment of distance effects on effective propagule flow among plant populations, separately for seeds and pollen. The method requires population genotypic data from biparentally inherited codominant genetic markers and a matrix of pairwise distances among populations. Although we have focused on geographical distance, the statistical framework could be easily adapted to assess the association between other types of ecological distance (e.g. altitudinal, phenological, wind connectivity, or landscape connectivity) and recent seed and pollen migration rates. The model also provides estimates of individual maternal and paternal ancestries, individual inbreeding coefficients, population allelic frequencies, per-locus allelic dropout rates, and levels of population genetic divergence. Our inference model should be of interest to ecologists and natural resource managers assessing how current landscape configuration influence, for instance, the reproductive and demographic connectivity of plant population isolates, the amount and geographical origin of long-distance migrants, or the introgression of allochthonous genes from plantations into natural populations. It is worth noting that our method would allow similar inference in animal species with dispersal syndromes analogous to plants. For instance, many sessile marine invertebrates (e.g. most sponges and some corals and mollusks) disperse sperm to fertilize retained eggs (Bishop and Pemberton 2006), subsequently dispersing larvae, and our approach could be used to analyze recent migration in such species, explicitly distinguishing both vectors of population connectivity.

The results of our simulation study suggest that the method can provide reliable estimates of seed and pollen migration rates and allow accurate inference of spatial effects on migration, at affordable sample sizes (between 25 and 50 individuals/population) when population genetic divergence is not low ($F_{ST} \geq 0.05$), or by increasing sampling (to at least 100 individuals/population) under weaker levels of divergence ($F_{ST} = 0.025$). An increasing number of

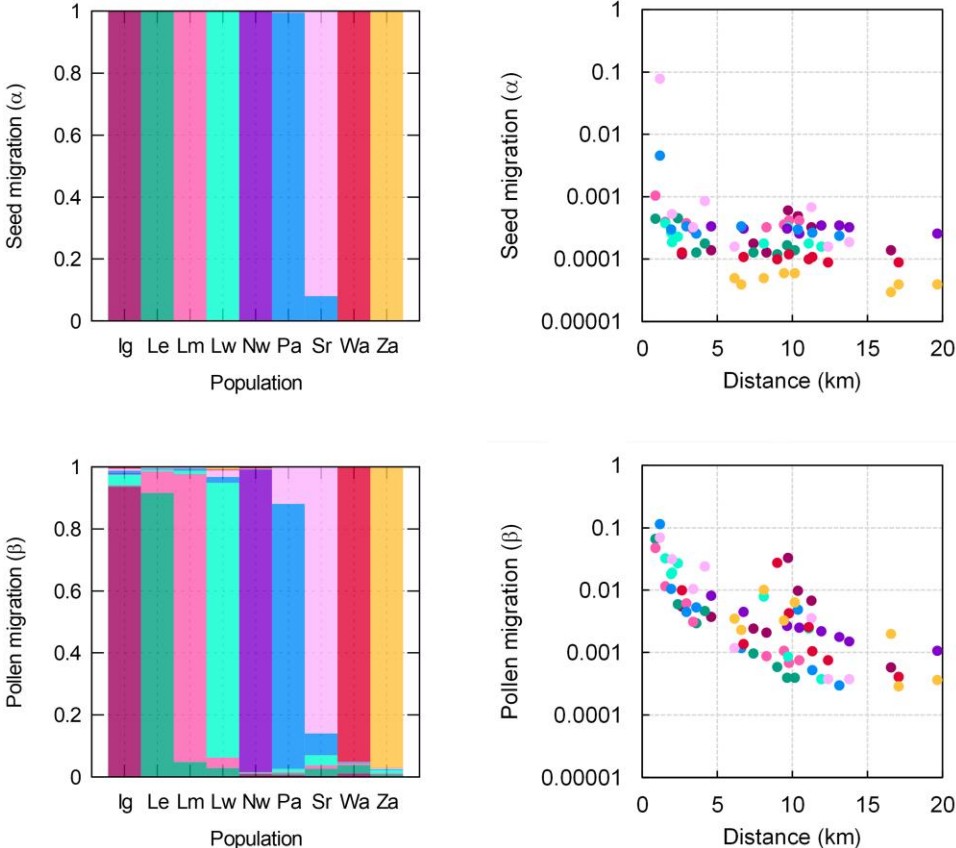

**Fig. 4.** Posterior estimates of seed and pollen migration rates among nine *T. baccata* remnant populations. Vertical bars in the left panels correspond to different local populations, with colors indicating the estimated proportions of seed or pollen from different population origins (the dominant color in each bar corresponds to the local population). Right panels show estimated pairwise migration rates as a function of interpopulation distance, with colors distinguishing estimates for each recipient population (as indicated by the dominant colors in the bar plots).

populations within the study region should not compromise inference relative to scenarios with less populations and the same individual global sample size, as it provides additional spatial information that helps improve posterior estimation of interpopulation distance effects on migration, provided per-population sample size does not become too low. In practice, given that no candidate source population within the study region should be left unsampled, per-population sample sizes should be increased as much as the total sampling budget allows. Estimation accuracy will generally be larger for seed than for pollen migration parameters, especially for weak population differentiation and small samples, a likely consequence of the fact that seed (zygotic) migrants generally carry twice as much allelic information from their maternal sources (unless they were in turn sired by migrant pollen) than any migrating pollen gamete does from its paternal one (Robledo-Arnuncio and Gaggiotti 2017). Our simulations also suggest that the method's genotyping intensity requirements will in practice be less stringent than the population genetic divergence ones, as the accuracy provided by assays with about 1,000 unlinked polymorphic SNP loci may approach, for a given sample size, the theoretical maximum achievable under categorical origin discrimination. It is also encouraging that the power to detect isolation by distance remained high (especially for seed migration) in all cases except for the unfavorable combination of small samples and weak population divergence, while the false positive rate remained conveniently low in general. Future studies could extend the range of demographic and sampling scenarios considered in our

simulations, to consider for instance the effect of linked loci, anisotropic migration, or varying levels of nonmigration.

We have applied our method to a previously published microsatellite data set for *T. baccata*, with a sample size, genotyping intensity, and level of population genetic divergence that, according to our simulation results, should allow accurate inference of recent seed and pollen migration and of spatial effects. Results revealed overall low seed and pollen migration between *T. baccata* populations during the last generation, with pairwise migration rates that were statistically significant for two pairs of nearby populations and with a negative effect of interpopulation distance on migration that was detectable for pollen but not for seeds. These results complement the low estimates of contemporary migration obtained for seedling samples from the same populations (Chybicki *et al.* 2024), suggesting that connectivity among the studied *T. baccata* remnants via seed and pollen migration was as low during the last generation as it presently is and that sustained fragmentation, as measured by long interpopulation distances, is hampering seed (and possibly pollen) exchange.

Characterizing isolation by distance patterns in recent migration can provide complementary information to the one provided by the observed distribution of migration rates itself. Analogously to the probability density function of dispersal distances from an individual plant that is inferred from the empirical distribution of dispersal events (Nathan *et al.* 2012), our model parameterization allows estimating a probability density function of migration distances from a population source based on the observed

distribution of migration rates (Supplement A in Supplementary File 1), providing a population-based description of average seed or pollen migration probabilities at different distances, including local dispersal. Such a statistical and modeling tool could be employed, for instance, to help make predictions about population connectivity levels via seed or pollen migration under alternative hypothetical spatial distribution scenarios considered for conservation management, to compare propagule migration potential among different plant taxa, to assess the risk of exotic introgression in different possible translocation scenarios, to parameterize plant metapopulation models, or, more generally, to build predictive or inference models that need to incorporate effective seed and pollen exchange among discrete plant populations in explicit space. We assumed a simple fat-tailed kernel in our model, the shape and scale of which are controlled by a single parameter (Supplementary A in Supplementary File 1). The choice was motivated by its statistical simplicity, considering that the number of populations, and therefore, the available spatial information may be limited in practice and by the fact that migration is mediated by long-distance propagule dispersal events that are best described using fat-tailed distributions (Klein, Lavigne, Gouyon, et al. 2006; Klein, Lavigne, Picault, et al. 2006). It would be possible to integrate more complex and flexible migration kernels into our model, e.g. with a second parameter to adjust the tail independently of the scale. No matter the assumed kernel, however, the method remains informative about the spatial patterns of migration when migration occurs mostly between nearest-neighbor populations (as with seed migration in our yew example), because in that case, isolation by distance is deduced directly from the distribution of migration rates rather than from the spatial parameter itself.

Our method complements existing ones for estimating recent migration rates based on population multilocus genotypic samples. It resembles BIMr (Faubet and Gaggiotti 2008) in that both consider an F-model for population allelic frequencies (Balding and Nichols 1997) and in that both focus on migration during the last generation, not distinguishing between first- and second-generation migrants, a simplification that allows considering migrants originating from two different populations and migration rates to vary between 0 and 1 (Faubet and Gaggiotti 2008). By contrast, BAYESASS (Wilson and Rannala 2003) does distinguish between F1 and F2 migrants, but at the cost of assuming single migrant ancestry and low migration rates (between 0 and 1/3). Besides estimating migration rates, both BIMr and our method additionally allow identifying factors affecting observed migration, by using a Dirichlet prior for migration rates with parameters that incorporate the environmental data (Gaggiotti et al. 2004). In our case, distance effects are incorporated into the prior using a "competing sources" framework (7), which has long been used to model relative propagule contributions from sources at different distances, both within (Adams et al. 1992; Oddou-Muratorio et al. 2005; Burczyk et al. 2006; Robledo-Arnuncio and García 2007) and among (Klein, Lavigne, Gouyon, et al. 2006; Klein, Lavigne, Picault, et al. 2006; Devaux et al. 2007) plant populations. The main addition of our method relative to both BIMr and BAYESASS is, however, that it explicitly considers both zygotic (seed) and gametic (pollen) migration and that it allows inferring the separate effect of spatial factors on each type of dispersal, providing a more suitable tool for plant species. The ESPM model also provides joint estimates of seed and pollen migration rates (Robledo-Arnuncio 2012; Robledo-Arnuncio and Gaggiotti 2017), but it does not include migration covariates. Another difference is that ESPM requires collecting samples before (adults) and after (offspring) one or several reference dispersal periods for which migration rates are estimated, whereas our model considers a single temporal sample and estimates migration during the last generation. It should finally be noted that even though dissecting zygotic and gametic components of migration will generally be of greatest interest in plant studies, the two components can be easily translated into total gene migration estimates for particular population genetic inferences or to be compared with estimates from other methods (see Supplement D in Supplementary File 1).

Although our method focuses on migration among populations, it also fills a gap in model-based inference of individual inbreeding, especially if allelic dropout or analogous issues (e.g. null alleles) cannot be neglected. The model that we used for inbreeding is similar to that implemented in INEST software (Chybicki and Burczyk 2009), except that we explicitly account for differences in allele frequencies between distinct sampling locations (i.e. subpopulations), thereby avoiding inbreeding overestimation due to the Wahlund effect. Our method can therefore be considered as an alternative to INEST when data from several populations are to be analyzed together. Moreover, our method provides a statistical framework that could be used for modeling population and individual factors of inbreeding, analogous to approaches developed for self-fertilization rates (Chybicki et al. 2019).

There are several limitations and possible extensions to our model. As most methods for estimating migration rates based on genotypic data, we assumed that all source populations have been sampled. Migration from unsampled sources tends to inflate migration rate estimates from sampled ones (Robledo-Arnuncio 2012), and there are theoretical approaches available to tackle this problem (Dawson and Belkhir 2001; Pella and Masuda 2006) that could be implemented and tested. Genotyping errors other than allelic dropouts, such as mistyping, can potentially bias migration estimates and could also be added to the model (Robledo-Arnuncio and Gaggiotti 2017). Regarding inferences on covariates of migration, we have only considered interpopulation distance, while multiple environmental factors could be simultaneously included into the prior of migration rates (Faubet and Gaggiotti 2008). For instance, population-specific factors that might affect the intensity of seed and pollen emission, such as size, density, or fecundity, could be directly incorporated into equation (7). It might also be possible to model local environmental factors influencing the probability that populations are isolated from immigrants (our $\tau$ parameter), such as physical barriers blocking propagule immigration (e.g. canopy closure, in the case of subcanopy species) or environmental factors driving selection against or for immigrants. Finally, an interesting alternative approach to investigate the influence of environmental factors on migration would be the association of a mechanistic model of wind- or animal-driven propagule migration to the statistical migration model based on genotypic likelihoods, which has been envisioned as a promising combination of methods (Kremer et al. 2012; Robledo-Arnuncio et al. 2014; Gaggiotti 2017) but remains largely unused.

As the speed of contemporary landscape and environmental changes increases, so are concerns about their impact on ecosystem functioning and services. Statistical tools are required to monitor the potentially complex interactions between these changes and fundamental biological processes, such as population reproductive and demographic connectivity. The method presented here advances in this direction, introducing a way to assess spatial and environmental covariates of effective seed and pollen exchange among plant populations.

## Data availability

Supplementary File 1 contains all supplemental material. A software package including the source code implementing the proposed model and used to generate the simulated data, as well as an executable binary file, test data, and a user manual, is available at https://doi.org/10.5281/zenodo.14474856.

Supplemental material available at GENETICS online.

## Acknowledgments

We thank the editor and an anonymous reviewer for their constructive suggestions.

## Funding

I.J.C. was supported by the National Science Centre, Poland (grant UMO-2018/31/B/NZ8/01808). J.J.R.-A. was supported by the Ministry for Ecological Transition and Demographic Challenge, Spain (grant MITECO2023-AF. 20234TE003).

## Conflicts of interest

The authors declare no conflicts of interest.

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

*Editor: G. Coop*