## [Peer Review File · Genetics]

Spatially explicit estimation of recent migration rates in plants using genotypic data

Igor Chybicki and Juan Robledo-Arnuncio

NOTE: The reviews and decision letters are unedited and appear as submitted by the reviewers.

In extremely rare instances and as determined by a Senior Editor or the EIC, portions of a review may be redacted. If a review is signed, the reviewer has agreed to no longer remain anonymous.

The review history appears in chronological order.

Review Timeline:

Submission Date:	2024-08-26
Editorial Decision:	2024-11-12
Resubmission Received:	2024-11-29
Editorial Decision:	2024-12-13
Revision Received:	2024-12-17
Accepted:	2024-12-21

November 12, 2024

GENETICS-2024-307404

Spatially explicit estimation of recent migration rates in plants using genotypic data

Dear Dr. Robledo-Arnuncio:

The reviewer and I have read your paper and appreciate its approach to the inference of migration. While your manuscript is not currently acceptable for publication in GENETICS, we would welcome a substantially revised manuscript. The reviewer's comments and concerns are listed at the end of this email and should to be addressed in a revised manuscript.

I agree with the reviewer that more simulations of different sampling strategies (small number of individuals) would be an important addition to the paper. This would allow readers and users to better understand the limitations and advantages of the method. We look forward to receiving your revised manuscript. Please let the editorial office know approximately how long you expect to need for revisions.

Upon resubmission, please include:

1. A clean version of your manuscript;
2. A marked version of your manuscript in which you highlight significant revisions carried out in response to the major points raised by the editor/reviewers (track changes is acceptable if preferred);
3. A detailed response to the editor's/reviewers' feedback and to the concerns listed above. Please reference line numbers in this response to aid the editor and reviewers.

Your paper will likely be sent back out for review.

Additionally, please ensure that your resubmission is formatted for GENETICS

<https://academic.oup.com/genetics/pages/general-instructions>

Follow this link to submit the revised manuscript: Link Not Available

Sincerely,

Graham Coop
Associate Editor
GENETICS

Approved by:
Nicholas Barton
Senior Editor
GENETICS

Reviewer #2 :

Chybicki and Robledo-Arnuncio present a new method for inferring migration rates from genomic data. Their method is applicable to both microsatellite and SNP data. I was pleased to see the authors make their method available for both types of sequencing data as many studies still use microsats - particularly in understudied species. The authors report that their method is capable of estimating recent migration rates with high accuracy and low bias - particularly with modest levels of genetic differentiation ($F_{st} > 0.05$). The paper is well written with informative figures. The open source software implementing the method the authors make available and is extremely well documented (the manual accompanying the method is crystal clear).

My biggest question is around the applicability of the method for other sampling schemes. The sample sizes required per population to get reliable estimates are rather large (≥ 50 individuals per population). The smallest population sample size that was tested was 25/population, correct? In the simulations exploring distance effects, the authors modelled 9 populations. This was presumably done to have results that compare with the yew data (as implied on Line 449). However, researchers may sample many more than this in some cases. The authors state in the Discussion that this is a topic for future study, but it really seems as if it would be appropriate in the present manuscript. Particularly, looking at sampling schemes containing more populations, but fewer individuals in each. I think that if the method were effective in such cases it would be of broader utility.

The above question is obviously related to how researchers assign individuals to populations. Would it matter if individuals were erroneously assigned to populations, or whether a handful of individuals represented distinct populations? Would that be effectively handled with the F statistics?

I think that the authors connection of inferred probability density functions to practical cases of conservation is very useful - particularly as it relates to assisted gene flow, a management technique that has been suggested for many tree species.

One other small thing: It would be helpful to clarify in the legends or the footer to Tables 1-3 and S1-S4 that N refers to the total sample size, rather than the number of individuals per population.

(NOTE: the line numbers mentioned below refer to the marked version of the manuscript)

EDITOR COMMENTS

COMMENT: *The reviewer and I have read your paper and appreciate its approach to the inference of migration. While your manuscript is not currently acceptable for publication in GENETICS, we would welcome a substantially revised manuscript. The reviewer's comments and concerns are listed at the end of this email and should to be addressed in a revised manuscript.*

RESPONSE: thank you for your time and constructive suggestions. We have addressed the reviewer's comments in the attached revision of the manuscript. Please find below itemized responses to specific comments.

COMMENT: *I agree with the reviewer that more simulations of different sampling strategies (small number of individuals) would be an important addition to the paper. This would allow readers and users to better understand the limitations and advantages of the method.*

RESPONSE: we have conducted additional simulations following the reviewer's suggestion. Specifically, as described below, we have added new scenarios enlarging the number of simulated populations (up to $K=20$) and decreasing the number of sampled individuals per population (down to a mean minimum of 12.5 individuals/pop). We believe the latter already represents a limit below which undertaking inference using our model should not be encouraged, as explained below.

REVIEWER #2 COMMENTS

COMMENT: *Chybicki and Robledo-Arnuncio present a new method for inferring migration rates from genomic data. Their method is applicable to both microsatellite and SNP data. I was pleased to see the authors make their method available for both types of sequencing data as many studies still use microsats - particularly in understudied species. The authors report that their method is capable of estimating recent migration rates with high accuracy and low bias - particularly with modest levels of genetic differentiation ($F_{st} > 0.05$). The paper is well written with informative figures. The open source software implementing the method the authors make available and is extremely well documented (the manual accompanying the method is crystal clear).*

RESPONSE: thanks for the positive assessment. We agree that considering SSR-type markers is still relevant when developing and testing migration, dispersal or gene flow inference methods.

COMMENT: *My biggest question is around the applicability of the method for other sampling schemes. The sample sizes required per population to get reliable estimates are rather large (≥ 50 individuals per population). The smallest population sample size that was tested was 25/population, correct? In the simulations exploring distance effects, the authors modelled 9 populations. This was presumably done to have results that compare with the yew data (as implied on Line 449). However, researchers may sample many more than this in some cases. The authors state in the Discussion that this is a topic for future study, but it really seems as if it*

would be appropriate in the present manuscript. Particularly, looking at sampling schemes containing more populations, but fewer individuals in each. I think that if the method were effective in such cases it would be of broader utility.

RESPONSE: we appreciate this relevant suggestion, the number of populations is certainly important when inferring migration, especially when using spatially-explicit methods. Yes, the previous minimum per-population sample size that we simulated was 25 individuals. We have now added 9 additional simulation scenarios following the reviewer's indications. In particular, we have now doubled the maximum number of populations considered, namely from $K=10$ to 20, or from $K=9$ to 18 in the case of the simulations exploring distance effects. In these new scenarios, the minimum per-population sample size considered has been reduced from an average of 25 to an average of 12.5 ($=250/20$). This was done assuming an intermediate level of population genetic divergence ($F_{st}=0.1$) and weak isolation by distance, when present. The new results show that increasing the number of simulated populations does not have much influence on the bias and RMSE of migration rate estimates when distance effects do not exist and are ignored. When distance effects are present, then increasing the number of simulated populations at constant total sample size does in fact reduce the estimation errors of both migration rates and distance effect parameters. Our interpretation is that increasing the number of populations provides additional spatial data for posterior inference that compensates the per-population data loss. However, there seems to be a minimum sample size per population below which inference worsens, especially in the case of pollen migration. It is difficult to provide general sampling rules without risking misguidance, as there are many factors involved, but we believe it would be risky to deliver the potentially over-optimistic message that accurate contemporary migration inference is feasible in general using less than 25-50 genotyped individuals per population. It also must be stressed that the number of sampled populations is not an adjustable parameter during sampling design: the method, as most of its kind, requires exhaustive sampling of candidate source populations. Therefore our recommendation is that, given the number of populations that is necessary to sample, one should increase the per-population sample size as much as the total sampling budget allows, at least to 25-50 individuals/pop.

All these results and comments have been added to the revised manuscript. In particular, in the methods (L222-231 and L245-250), results (L326-333 and L400-408 and Tables E and H in Supplementary File S1) and discussion (L459-469). We have chosen to include the new Tables as supplementary material, given that the main text already contains a substantial amount of (big) tables and figures.

COMMENT: *The above question is obviously related to how researchers assign individuals to populations. Would it matter if individuals were erroneously assigned to populations, or whether a handful of individuals represented distinct populations? Would that be effectively handled with the F statistics?*

RESPONSE: our method, as others based on genetic assignment of individuals to populations, is adequate for discrete populations only. So, if we understand the comment correctly, there should be no possible choice or error in the allocation of individuals to sampling origins (i.e. populations), other than mislabeling or similar low-frequency mistakes that should in principle be of minor concern. If the reviewer rather refers to wrong posterior statistical assignment of individuals to the source population before migration, then the answer would be that the simulation results indicate that the probabilistic assignment is generally accurate enough to produce unbiased estimates of migration rates, unless population genetic divergence and/or sample size is too low, as discussed in the manuscript.

Concerning the possibility of a handful of individuals representing distinct populations: if this refers to individuals migrating from unsampled populations, then this possibility is discussed in lines 558-562. If the reviewer rather means the possibility that a sampled population does actually comprise two distinct gene pools within its boundaries, then this should not be a problem for our model, as the estimated recent migration rates always refer to the sampled contemporary candidate source populations, independently of past demographic history before migration. But yes, this kind of specific genetic structure patterns would anyway be properly reflected in the estimated population allelic frequencies and F_{st} values.

COMMENT: *I think that the authors connection of inferred probability density functions to practical cases of conservation is very useful - particularly as it relates to assisted gene flow, a management technique that has been suggested for many tree species.*

RESPONSE: indeed, we agree that inferring a migration dispersal kernel is a potentially useful tool for interesting applications in conservation, such as the one mentioned by the reviewer. The software outputs the estimated migration kernel parameters for this and other potential purposes.

Given the potential practical utility of the estimated migration kernel, and its relative novelty, we have added a comment in the discussion (L509-519) elaborating a bit more on the shape of the assumed distribution (and the possibility of incorporating alternatives), and how it can be expected to be more sensitive to long-distance migration events. We believe this comment may help readers understand the utility of the spatial model in detecting connectivity across a landscape. This addition may also help interpret the seed migration results in our yew case study, namely that seed migration occurred between nearest neighbor populations only and that the spatial model had low statistical support.

COMMENT: *One other small thing: It would be helpful to clarify in the legends or the footer to Tables 1-3 and S1-S4 that N refers to the total sample size, rather than the number of individuals per population.*

RESPONSE: we have specified that N refers to the total sample size in all tables and figures, thanks.

December 13, 2024
RE: GENETICS-2024-307668

Dear Dr. Robledo-Arnuncio:

I am pleased to accept your manuscript entitled "Spatially explicit estimation of recent migration rates in plants using genotypic data" for publication in GENETICS, pending minor revision. I thank you for all your hard work in response to the review process. The paper currently states that the code will be uploaded upon acceptance. Please add links to the code to the manuscript. I expect you should be able to submit a revised manuscript within 30 days. A suitably revised manuscript will be acceptable for publication; I don't expect to send it out for review.

Please ensure that you have included a Data Availability Statement at the end of the Materials and Methods section. Details available at <https://academic.oup.com/genetics/content/prep-manuscript>. The DAS should include the accession numbers or DOIs of any data you have placed in public repositories, describe supplemental material, include applicable IRB numbers, and may include specifications for how to properly acknowledge or cite the data.

When revising the ms., please make an effort to shorten it, because that almost always improves a manuscript. We urge authors to heed the advice of Strunk and White: "omit needless words"¹. Follow this link to submit the revised manuscript: Link Not Available

Thank you for submitting this story to Genetics.

Sincerely,

Graham Coop
Associate Editor
GENETICS

Approved by:
Nicholas Barton
Senior Editor
GENETICS

EDITOR COMMENTS

COMMENT: *I am pleased to accept your manuscript entitled "Spatially explicit estimation of recent migration rates in plants using genotypic data" for publication in GENETICS, pending minor revision. I thank you for all your hard work in response to the review process. The paper currently states that the code will be uploaded upon acceptance. Please add links to the code to the manuscript.*

RESPONSE: thank you for your time and effort to improve our manuscript. We have now added to the Data Availability Statement a link to a public repository entry where the presented software package is freely available for download. The package contains the code used in the manuscript as well as an executable binary file, test data and a user manual.

COMMENT: *Please ensure that you have included a Data Availability Statement at the end of the Materials and Methods section. The DAS should include the accession numbers or DOIs of any data you have placed in public repositories, describe supplemental material, include applicable IRB numbers, and may include specifications for how to properly acknowledge or cite the data.*

RESPONSE: the DAS has been updated to include the Zenodo's link-DOI of the software package. We have also included the following sentence: "File S1 contains all supplemental material", and we have updated File S1 to include an initial content section summarizing all the material included in the file.

COMMENT: *When revising the ms., please make an effort to shorten it, because that almost always improves a manuscript.*

RESPONSE: we agree and appreciate the suggestion. We had already made a big effort to shorten the manuscript before the initial submission and during the previous round of review. For instance, we moved a lot of important content to the appendixes included in supplemental File S1, and we tried to make the main text (ca. 7000 words long) as succinct as possible without losing clarity. We believe shortening it significantly further would result in undesirable conceptual trimming.

December 17, 2024

RE: GENETICS-2024-307668R1

Dr. Juan J Robledo-Arnuncio
Instituto Nacional de Investigacion y Tecnologia Agraria y Alimentaria
Instituto de Ciencias Forestales (ICIFOR-INIA)
Ctra. de la Coruña km 7.5
Madrid 28040
Spain

Dear Dr. Robledo-Arnuncio:

Congratulations! We are delighted to inform you that your manuscript entitled "Spatially explicit estimation of recent migration rates in plants using genotypic data" is acceptable for publication in GENETICS. Many thanks for submitting your research to the journal.

To Proceed to Production:

1. Format your article according to GENETICS style, as discussed at <https://academic.oup.com/genetics/pages/general-instructions>, and upload your final files at <https://genetics.msubmit.net>.
2. Your manuscript will be published as-is (unedited-as submitted, reviewed, and accepted) at the GENETICS website as an Advanced Access article and deposited into PubMed shortly after receipt of source files and the completed license to publish. Please notify sourcefiles@thegsajournals.org if you do not wish to publish your article via Advanced Access.
3. We invite you to submit an original color figure related to your paper for consideration as cover art. Please email your submission to the editorial office or upload it with your final files. You can submit a small-sized image for evaluation, and if selected, the final image must be a TIFF file 2513px wide by 3263px high (8.375 by 10.875 inches; resolution of 600ppi). Please avoid graphs and small type.

If you have any questions or encounter any problems while uploading your accepted manuscript files, please email the editorial office at sourcefiles@thegsajournals.org.

Sincerely,

Graham Coop
Associate Editor
GENETICS

Approved by:
Nicholas Barton
Senior Editor
GENETICS

note: Please add jnls.author.support@oup.com and genetics.oup@kwglobal.com (or the domains @oup.com and @kwglobal.com) to your email program's "safe senders" list. You will be contacted by both at various points during the production process.

Review comments (if applicable):